# Can prognostic factors for indirect muscle injuries in elite football (soccer) players be identified using data from preseason screening? An exploratory analysis using routinely collected periodic health examination records

Tom Hughes  ,[1,2,3] Richard Riley  ,[4] Michael J Callaghan,[1,2,3] Jamie C Sergeant[2,5]

MJC and JCS contributed equally.

For numbered affiliations see end of article.

**Correspondence to**
Dr Tom Hughes;
tom.hughes.physio@manutd.co.uk

## ABSTRACT

**Background** In elite football, periodic health examination (PHE) may be useful for injury risk prediction.

**Objective** To explore whether PHE-derived variables are prognostic factors for indirect muscle injuries (IMIs) in elite players.

**Design** Retrospective cohort study.

**Setting** An English Premier League football club.

**Participants** 134 outfield elite male players, over 5 seasons (1 July 2013–19 May 2018).

**Outcome and analysis** The outcome was any time-loss, lower extremity index IMI (I-IMI). Prognostic associations were estimated using odds ratios (ORs) and corresponding statistical significance for 36 variables, derived from univariable and multivariable logistic regression models. Missing data were handled using multiple imputation. Non-linear associations were explored using fractional polynomials.

**Results** During 317 participant-seasons, 138 I-IMIs were recorded. Univariable associations were determined for previous calf IMI frequency (OR 1.80, 95% CI 1.09 to 2.97), hamstring IMI frequency (OR 1.56, 95% CI 1.17 to 2.09), if the most recent hamstring IMI occurred >12 months but <3 years prior to PHE (OR 2.95, 95% CI 1.51 to 5.73) and age (OR 1.12 per 1-year increase, 95% CI 1.06 to 1.18). Multivariable analyses showed that if a player's most recent previous hamstring IMI was >12 months but <3 years prior to PHE (OR 2.24, 95% CI 1.11 to 4.53), this was the only variable with added prognostic value over and above age, which was a confirmed prognostic factor (OR 1.12 per 1-year increase, 95% CI 1.05 to 1.18). Allowing non-linear associations conferred no advantage over linear associations.

**Conclusion** PHE has limited use for injury risk prediction. Most variables did not add prognostic value over and above age, other than if a player experienced a hamstring IMI >12 months but <3 years prior to PHE. However, the precision of this prognostic association should be confirmed in future.

**Trial registration number** NCT03782389.

## STRENGTHS AND LIMITATIONS OF THIS STUDY

⇒ This study investigated a wide selection of periodic health examination-derived variables and their association with index indirect muscle injuries in a cohort of elite football players, over and above standard anthropometric variables of age, height and weight.

⇒ This is the first known study in elite football to explore any non-linear associations between injury outcomes and variables measured on a continuous scale, using a fractional polynomial approach.

⇒ High-quality cohort data were used, with variables reliably measured in preseason. Any subsequent injury outcomes were confirmed using a validated muscle injury classification system.

⇒ Some participants had missing data for some factors; a multiple imputation approach was used to help address this, under a missing at random assumption.

⇒ Candidate factors were only measured at one time-point each season, which means that dynamic associations were not investigated.

## BACKGROUND

Periodic health examination (PHE), or screening, is a well-established clinical evaluation strategy in elite football.[1] Typically during PHE, players undertake various medical, musculoskeletal, functional and performance tests[2] during preseason and in-season periods.[1] PHE allows opportunities for general health surveillance, identification of salient pathology[3] and monitoring of rehabilitation or performance.[4] In addition, although it is unlikely that PHE can establish specific causal factors for injuries,[4] it is perceived to be useful for the prediction of future injury risk in athletes,[2 4] which could

prove especially valuable for injury types that are problematic in terms of incidence and severity. Indirect (non-contact) muscle injuries (IMIs) are an obvious example, because they account for between 30.3% and 47.9% of all injuries observed in elite football[5–9] and each IMI typically results in 14.4[5] to 15 days lost to training and competition.[8]

To be able to predict the risk of future health events, prognostic factors are required.[10] In the context of football, prognostic factors could be any PHE derived variables, characteristics or measurements (eg, medical history, leg strength or range of motion tests) that are associated with increased injury risk through causal or non-causal pathways.[4] There is clinical value in gaining a deeper understanding of prognostic factors associated with injuries such as IMIs. Specifically, prognostic factors can help practitioners understand the differences in risk (outcome event probability) between players, and therefore, explain why some players may have a better or worse prognosis than others.[4] Furthermore, prognostic factors that have an established causal role in injury occurrence can inform the selection of injury mitigation strategies, relevant for subgroups of players who share such characteristics.[4] Finally, causal prognostic factors can also be used to develop innovative intervention approaches aimed at mitigating risk.[10]

Despite these benefits, individual prognostic factors have limited predictive power.[11 12] However, if several prognostic factors are used in combination within a multivariable prognostic model, it may be possible to produce useful individualised risk estimates[10 11] that can be used to communicate risks to practitioners and coaches.[13] Additionally, if developed using prognostic factors which have a causal role in injury risk, prognostic models could also be used to assist practitioners in selecting an array of specific risk reduction interventions that are bespoke to the prognostic factor profile of individual players.[4]

Because the predictive function of PHE remains unsubstantiated[3 14] and given that IMIs are the most significant problem observed in elite football,[5–9] a related multivariable prognostic model was recently developed to predict individualised lower extremity IMI risk in elite players using PHE data.[13] However, sample size limitations meant that only 10 candidate prognostic factors could be considered in the model and these were selected using data quality assessment, clinical reasoning, or on the basis of a related systematic review.[14] The performance of the model was modest and it was concluded that implementing it in practice would not be beneficial.[13]

Furthermore, several methodological limitations of the current evidence have been previously highlighted, which specifically included inadequate reporting of outcomes, prognostic factor measurement and reliability.[14] Additionally, while most studies performed appropriate statistical analyses, continuous prognostic factor measurements were often categorised[15–18] and non-linear associations were not investigated,[15–21] which does not conform to current methodological recommendations.[22–24]

To further the development of IMI prognostic models and improve understanding of how differences in IMI risk may occur between individuals, there is a clear need to ascertain the existence of other robust and novel prognostic factors.[13] Therefore, this study used routinely collected data from a five-season period to explore: (1) prognostic associations between PHE-derived data and IMI outcomes in elite footballers, using a broader dataset than had been considered in the development of the previous prognostic model[13] and (2) the prognostic value of these PHE-derived data over and above standard anthropometric data, including age (which has previously confirmed prognostic value[13 14]), height and weight. Both linear and non-linear associations were also explored, which, as far as is known, has not been conducted previously.

## METHODS

The methodology has been described in a published protocol[25] so will only be briefly outlined. This study was registered on ClinicalTrials.gov (NCT03782389) and was reported according to the Reporting Recommendations for Marker Prognostic Studies.[26] Given the number of PHE-related variables examined, our study should be viewed as exploratory, but we emphasise that this an important phase in prognostic factor research.[12 27]

### Data sources

This study was of retrospective cohort design. Eligible participants were identified from a population of male elite footballers, aged 16–40 years old at an English Premier League football club. A database was created by the principal investigator (TH) using routinely collected injury records and preseason PHE data over five seasons (1 July 2013–19 May 2018). This process included checks for accuracy, duplicate or missing entries. Participants completed a mandatory PHE during the first week of each season (which started on 1 July), and were followed up to the last first team game of the season.

The PHE process typically included: (1) anthropometric measurements; (2) a review of medical and previous injury history; (3) musculoskeletal examination tests; (4) functional movement and balance tests and (5) strength and power tests. Descriptions of all included test procedures are presented in online supplemental file 1. The PHE test order was self-selected by each player and a standardised warm-up was not implemented, although players could undertake their own warm-up procedures if they wished. Each component of PHE was standardised according to a written protocol and was examined by physiotherapists, sports scientists or club medical doctors. The same examiners performed the same test every season to eliminate intertester variability. No examiner attrition occurred throughout the data collection period. If a participant was injured at the scheduled time of PHE, a risk assessment was completed by medical staff and participants only completed tests that were deemed appropriate

and safe for the participant's condition; examiners were therefore not blinded to injury status.

## Eligibility criteria

During any season, participants were eligible if they: (1) were not a goalkeeper and (2) participated in PHE for the relevant season. Participants were excluded if they were not under contract to the club at the time of PHE.

## Patient and public involvement

Participants and members of the public were not involved in the study design.

## Outcome

The outcome was any time-loss, index lower extremity IMI (I-IMI) sustained by a participant during match play or training, which affected any lower abdominal, hip, thigh, calf or foot muscle groups and prohibited future match or training participation.[28] I-IMIs were confirmed and graded by a club doctor or physiotherapist according to the previously validated Munich Consensus Statement for the Classification of Muscle Injuries in Sport,[29 30] during routine assessments undertaken within 24 hours of injury occurrence. The medical professionals were not blinded to PHE data at diagnosis.

## Sample size

Our sample size of 317 participant-seasons (with 138 I-IMI events) had 80% power to detect an adjusted OR of at least 1.6 for a 1 SD increase in a variable of interest, conservatively assuming a correlation of 0.5 with the adjustment variables of age, height and weight (see online supplemental file 2 for the sample size calculation).[31]

## PHE-derived candidate variables

To aid clarity in this study, the term 'prognostic factor' is reserved for variables found to have a prognostic association with an I-IMI outcome (ie, with statistical evidence established during the analyses), whereas the term 'candidate variables' relates to all variables for which the association with I-IMI outcome was investigated during the analyses.

As described in the study protocol, the dataset contained 60 variables[25] that were eligible for analysis unless there were >15% missing observations or if reliability (where applicable) was reported as fair to poor (ie, intraclass correlation coefficient <0.70).[25 32] If any variables did not meet these eligibility criteria, they were excluded (online supplemental file 3). Collinearity between eligible variables was assessed with a scatterplot matrix; this was evident when tests were used to measure right and left limbs independently.[25] In these circumstances, composite variables were created for between-limb differences and the mean of the test measurements for both limbs, as described in the study protocol.[25]

Of the remaining eligible variables, 10 were used in a previous study to develop a multivariable prognostic model for I-IMI prediction (represented by 12 parameters).[13] With the exception of age at PHE (which was

used for adjustment purposes in this study), these candidates were therefore excluded.[25] The final number of candidate variables included for exploratory analysis was 36. Table 1 summarises all included variables with their measurement units and data type, as well as the measurement methods, their reliability and validity.

## Statistical analysis

### Data handling: outcome measures

Each participant-season was treated as independent. If an I-IMI occurred, the participant's outcome was determined for that season and they were no longer considered at risk. In these circumstances, participants were included for further analysis at the start of the consecutive season, if still eligible. Any upper limb IMI, trunk IMI or non-IMI injuries were ignored and participants were still considered at risk.

Eligible participants who were loaned to another club throughout that season, but had not sustained an I-IMI prior to the loan were still considered at risk. I-IMIs that occurred while on loan were included for analysis. Permanently transferred participants (who had not sustained an I-IMI prior to the transfer), were recorded as not having an I-IMI during the relevant season and exited the cohort at the season end.

### Data handling: missing data

Missing values were assumed to be missing at random (ie, missingness could be predicted conditional on other known variables).[25] The continuous parameters generally demonstrated non-normal distributions, so were transformed using normal scores[33] to approximate normality before imputation and back-transformed following imputation.[34] Multivariate normal multiple imputation was performed, using a model that included all candidate variables and I-IMI outcomes. Fifty imputed datasets were created in Stata V.15.1 (StataCorp) using the 'mi impute' command.

### Univariable and multivariable analyses

All data were analysed in the form that they were recorded. In particular, variables that were recorded as continuous were kept continuous and not categorised, to avoid a loss of prognostic information.[22] Univariable logistic regression models were used to estimate the unadjusted linear associations between I-IMIs and each candidate variable. Multivariable logistic regression models were also used to estimate the linear association between I-IMIs and each variable, after adjustment for age (which has confirmed prognostic importance[13 14]), height and weight (which were both considered as potential confounders for I-IMIs and PHE-derived candidates). All parameter estimates were averaged across all imputed datasets using Rubin's Rules[35] and were computed using the 'mim' module in Stata V.15.1. Statistical significance thresholds were used to indicate the strength of exploratory evidence against null associations, where p values of : (1) <0.05 indicated strong evidence and the variable was considered

**Table 1** List of candidate variables with a summary of the units of measurement, methods and reliability of measurement, and data type

| Candidate variable type | Name of candidate variable | Candidate variable identification no. | Measurement unit | Measurement method | Reliability (if applicable/ available) | Validity (if applicable/ available) | Data type |
|---|---|---|---|---|---|---|---|
| Anthropometric | Age | 1 | Years | Medical records | – | – | Cont. |
| | Height | 2 | Centimetres (cm) | Standing height measure | – | – | Cont. |
| | Weight | 3 | Kilograms (kg) | Digital scales | – | – | Cont. |
| Medical history | Within 3 years prior to PHE, the frequency of: Foot/ankle injuries | 4 | Count | Medical records | – | – | Dis./cont. |
| | Hip/groin injuries | 5 | Count | Medical records | – | – | Dis./cont. |
| | Knee injuries | 6 | Count | Medical records | – | – | Dis./cont. |
| | Shoulder injuries | 7 | Count | Medical records | – | – | Dis./cont. |
| | Lumbar spine injuries | 8 | Count | Medical records | – | – | Dis./cont. |
| | Iliopsoas injuries | 9 | Count | Medical records | – | – | Dis./cont. |
| | Hip adductor IMIs | 10 | Count | Medical records | – | – | Dis./cont. |
| | Hamstring IMIs | 11 | Count | Medical records | – | – | Dis./cont. |
| | Quadriceps IMIs | 12 | Count | Medical records | – | – | Dis./cont. |
| | Calf IMIs | 13 | Count | Medical records | – | – | Dis./cont. |
| | Within 3 years prior to PHE, the most recent Foot/ankle injury | 14 | Never, <6 months, 6–12 months, >12 months. | Medical records | – | – | Cat. |
| | Hip/groin injury | 15 | Never, <6 months, 6–12 months, >12 months. | Medical records | – | – | Cat. |
| | Knee injury | 16 | Never, <6 months, 6–12 months, >12 months. | Medical records | – | – | Cat. |
| | Shoulder injury | 17 | Never, <6 months, 6–12 months, >12 months. | Medical records | – | – | Cat. |
| | Lumbar spine injury | 18 | Never, <6 months, 6–12 months, >12 months. | Medical records | – | – | Cat. |
| | Iliopsoas injury | 19 | Never, <6 months, 6–12 months, >12 months. | Medical records | – | – | Cat. |
| | Hip adductor IMI | 20 | Never, <6 months, 6–12 months, >12 months. | Medical records | – | – | Cat. |
| | Hamstring IMI | 21 | Never, <6 months, 6–12 months, >12 months. | Medical records | – | – | Cat. |

Continued

**Table 1** Continued

| Candidate variable type | Name of candidate variable | Candidate variable identification no. | Measurement unit | Measurement method | Reliability (if applicable/ available) | Validity (if applicable/ available) | Data type |
|---|---|---|---|---|---|---|---|
| | Quadriceps IMI | 22 | Never, <6 months, 6–12 months, >12 months. | Medical records | – | – | Cat. |
| | Calf IMI | 23 | Never, <6 months, 6–12 months, >12 months. | Medical records | – | – | Cat. |
| Musculoskeletal | Mean PROM hip internal rotation | 24 | Degrees | Digital inclinometer+ROM | Intrarater ICC=0.90[42] | – | Cont. |
| | Mean PROM hip external rotation | 25 | Degrees | Digital inclinometer+ROM | Intrarater ICC=0.90[42] | – | Cont. |
| | Mean hip flexor length | 26 | Degrees | Digital inclinometer+Thomas Test | Inter-rater ICC=0.89[43] | Concurrent validity with handheld goniometer (r=0.86–0.92)[43] and IMC (r=0.49–0.53)[44] | Cont. |
| | Mean hamstring length/neural mobility | 27 | Degrees | Digital inclinometer+SLR | Intrarater ICC=0.95–0.98[45] Inter-rater ICC=0.80–0.97[45] | Construct validity with handheld inclinometer (r=0.98–0.99)[46] | Cont. |
| | Mean calf muscle length | 28 | Degrees | Digital inclinometer+WBL | Inter-rater ICC=0.80–0.95[47 48] Intrarater ICC=0.88[48] | Concurrent validity of inclinometer with 2D motion analysis (r range=0.71–0.76)[49] | Cont. |
| Strength/power | Max. leg extension power difference | 29 | Normalised watts per kilo (W/kg$^{-0.67}$) | Double leg press using Keiser Air 300 machine | Test–retest ICC=0.886[50] | Concurrent validity with mounted force plate (r=0.952)[51] | Cont. |
| | Mean of max. leg extension power | 30 | Normalised watts per kilo (W/kg$^{-0.67}$) | Double leg press using Keiser Air 300 machine | Test–retest ICC=0.886[50] | Concurrent validity with mounted force plate (r=0.977)[51] | Cont. |
| | Max. leg extension velocity difference | 31 | Peak velocity (m.s$^{-1}$) | Double leg press using Keiser Air 300 machine | Test–retest ICC=0.792[50] | Concurrent validity with mounted force plate (r=0.999)[51] | Cont. |
| | Mean of max. leg extension velocity | 32 | Peak velocity (m.s$^{-1}$) | Double leg press using Keiser Air 300 machine | Test–retest ICC=0.792[50] | Concurrent validity with mounted force plate (r=0.999)[51] | Cont. |
| | Max leg extension force difference | 33 | Normalised peak force (N/kg$^{-0.67}$) | Double leg press using Keiser Air 300 machine | Test–retest ICC=0.914[50] | Concurrent validity with mounted force plate (r=0.994)[51] | Cont. |
| | Mean of max. leg extension force | 34 | Normalised peak force (N/kg$^{-0.67}$) | Double leg press using Keiser Air 300 machine | Test–retest ICC=0.914[50] | Concurrent validity with mounted force plate (r=0.994)[51] | Cont. |

Continued

**Table 1** Continued

| Candidate variable type | Name of candidate variable | Candidate variable identification no. | Measurement unit | Measurement method | Reliability (if applicable/available) | Validity (if applicable/available) | Data type |
|---|---|---|---|---|---|---|---|
| | CMJ Force per kg of body mass | 35 | Force per kg (N/kg) | CMJ+force plate | Test–retest ICC=0.80–0.88[52] | Concurrent validity with other force plate (r≥0.99)[53] | Cont. |
| | CMJ height | 36 | Centimetres (cm) | CMJ+force plate | Test–retest ICC=0.80–0.88[52] | Concurrent validity with other force plate (r≥0.99)[53] | Cont. |

N (note: $N/kg^{-0.67}$ has a scaling factor to normalise force to body mass).[54] W (note: $W/kg^{-0.67}$ has a scaling factor to normalise force to body mass).[54] Cat., categorical; CMJ, countermovement jump; Cont., Continuous; dis./cont., discrete treated as continuous; ICC, intraclass correlation coefficient; I-IMI, index indirect muscle injury; IMC, inertial motion capture; IMI, indirect muscle injury; $m.s^{-1}$, metres per second; PHE, periodic health examination; PROM, passive range of movement; ROM, range of movement; SLR, straight leg raise; WBL, weight-bearing lunge.

significant; (2) 0.05–0.10 indicated weak evidence and (3) >0.10 indicated little or no evidence.[36] Prognostic importance was also considered by checking the magnitude of prognostic effects encompassed by the width of 95% CIs.

For all variables, non-linear associations with the outcome were also explored using fractional polynomials for the univariable and multivariable models; the fit of first and second order fractional polynomial models were evaluated against the fit of the standard logistic regression models.[37] The parameter estimates were combined across all imputed datasets[38] using Rubins Rules,[35] with the automated 'mfpmi' algorithm in Stata V.15.1, using a significance threshold set at p<0.05. All analyses are summarised in table 2.

### Primary and sensitivity analyses

To determine the effect of imputation and player transfers on variable associations, the analyses were repeated: (1) as complete cases analyses and (2) as sensitivity analyses excluding participant-seasons for participants who were loaned or transferred (performed as both multiple imputation and complete case analyses). All primary complete case and sensitivity analyses are also summarised in table 2.

### RESULTS

### Participants

During the five seasons, 134 participants were included, contributing 317 participant-seasons and 138 IMIs in the primary analysis (figure 1). Three players were classified as injured at the time of PHE (which affected three participant-seasons). This meant they were unavailable for training or match selection at that time. However, these players had commenced football specific, field-based rehabilitation and so had exposure to similar training activities to uninjured players. Therefore, they were included in the cohort because it was reasonable to assume that they could also be considered at risk of an I-IMI event.

For the sensitivity analyses (excluding loans and transfers), 260 independent participant-seasons with 129 IMIs were included; 36 participants were transferred on loan, while 14 participants were permanently transferred during a season, which excluded 57 participant-seasons (figure 1).

Table 3 summarises the participant characteristics and candidate variable values for participants included in the primary analyses. All values were similar to those included in the sensitivity analyses (online supplemental file 4).

### Missing data and multiple imputation

Data were complete for age and all past medical history variables (table 3). For all other candidates, the proportion of missing data ranged from 5.68% (for height and weight) to 14.20% (for the mean and between limb differences of maximal leg extension power and force)

**Table 2** Summary of all statistical analyses performed

| Statistical analyses of I-IMI outcomes | | | | | |
|---|---|---|---|---|---|
| Analysis performed | Participant-seasons | No of I-IMIs | Variables considered | Adjusted for | Results |
| Primary analysis, imputed data | | | | | |
| A1: Univariable logistic regression/FPs | 317 | 138 | Individual models for I-IMI+each variable (1–36) | None | table 4 |
| A2: Multivariable logistic regression/FPs | 317 | 138 | Individual models for I-IMI+each variable (4–36) | Variables 1–3 (age, height, weight) | table 4 |
| Primary analysis, complete case data | | | | | |
| B1: Univariable logistic regression/FPs | 265 | 115 | Individual models for I-IMI+each variable (1–36) | None | online supplemental file 6 |
| B2: Multivariable logistic regression/FPs | 265 | 115 | Individual models for I-IMI+each variable (4–36) | Variables 1–3 (age, height, weight) | online supplemental file 6 |
| Sensitivity analysis, imputed data | | | | | |
| C1: Univariable logistic regression/FPs | 260 | 129 | Individual models for I-IMI+each variable (1–36) | None | online supplemental file 7 |
| C2: Multivariable logistic regression/FPs | 260 | 129 | Individual models for I-IMI+each variable (4–36) | Variables 1–3 (age, height, weight) | online supplemental file 7 |
| Sensitivity analysis, complete case data | | | | | |
| D1: Univariable logistic regression/FPs | 217 | 106 | Individual models for I-IMI+each variable (1–36) | None | online supplemental file 8 |
| D2: Multivariable logistic regression/FPs | 217 | 106 | Individual models for I-IMI+each variable (4–36) | Variables 1–3 (age, height, weight) | online supplemental file 8 |

FP, fractional polynomials; I-IMI, index indirect muscle injury.

(table 3). For all continuous variables, the distribution of imputed values approximated the observed values (online supplemental file 5), therefore, confirming their plausibility.

### Univariable analyses

Table 4 shows the results of the univariable analyses. The continuous variables of age (OR 1.12 for a 1 year increase, 95% CI 1.06 to 1.18, p<0.001), weight (OR 1.03 for a 1 kg increase, 95% CI 1.00 to 1.07, p=0.03) and mean hip IR PROM (OR=0.97 for a 1 degree increase, 95% CI 0.95 to 0.99, p=0.01) showed a significant but modest association with I-IMIs. The narrow CIs indicated that these estimates were relatively precise. Linear associations were the best fit for all these continuous variables. Significant associations with larger OR estimates were observed for previous calf IMI frequency (OR 1.80, 95% CI 1.09 to 2.97, p=0.02), hamstring IMI frequency (OR 1.56, 95% CI 1.17 to 2.09, p<0.001), and if the most recent hamstring IMI occurred more than 12 months but less than 3 years prior to PHE (OR 2.95, 95% CI 1.51 to 5.73, p<0.001). The wider CIs for these estimates indicated greater imprecision about the prognostic effect; this may because these candidates were either discrete or categorical, rather than continuous.

Despite relatively large ORs, weaker evidence of associations was observed for the frequency of previous shoulder injuries (OR 2.38, 95% CI 0.98 to 5.75, p=0.05) and if the most recent calf IMI was less than 6 months prior to PHE (OR 3.78, 95% CI 0.98 to 14.56, p=0.05). However, the very wide CIs indicated considerable uncertainty about the true OR. No other significant candidate factors were observed.

### Multivariable analyses

Table 4 shows the results of the multivariable analyses, where the adjusted prognostic value was evaluated for all PHE-derived variables. After adjustment for height and weight, age remained significantly associated with increased odds of sustaining an I-IMI during a season (OR=1.12 for a 1-year increase, 95% CI=1.05 to 1.18, p<0.001) and a linear association was the best fit for this variable. However, there was no evidence that height and weight were strong prognostic factors independent of age.

After adjustment for age, height and weight, if the most recent hamstring IMI was more than 12 months but less than 3 years prior to PHE, the significant association and wide CI also remained (OR 2.24, 95% CI 1.11 to 4.53,

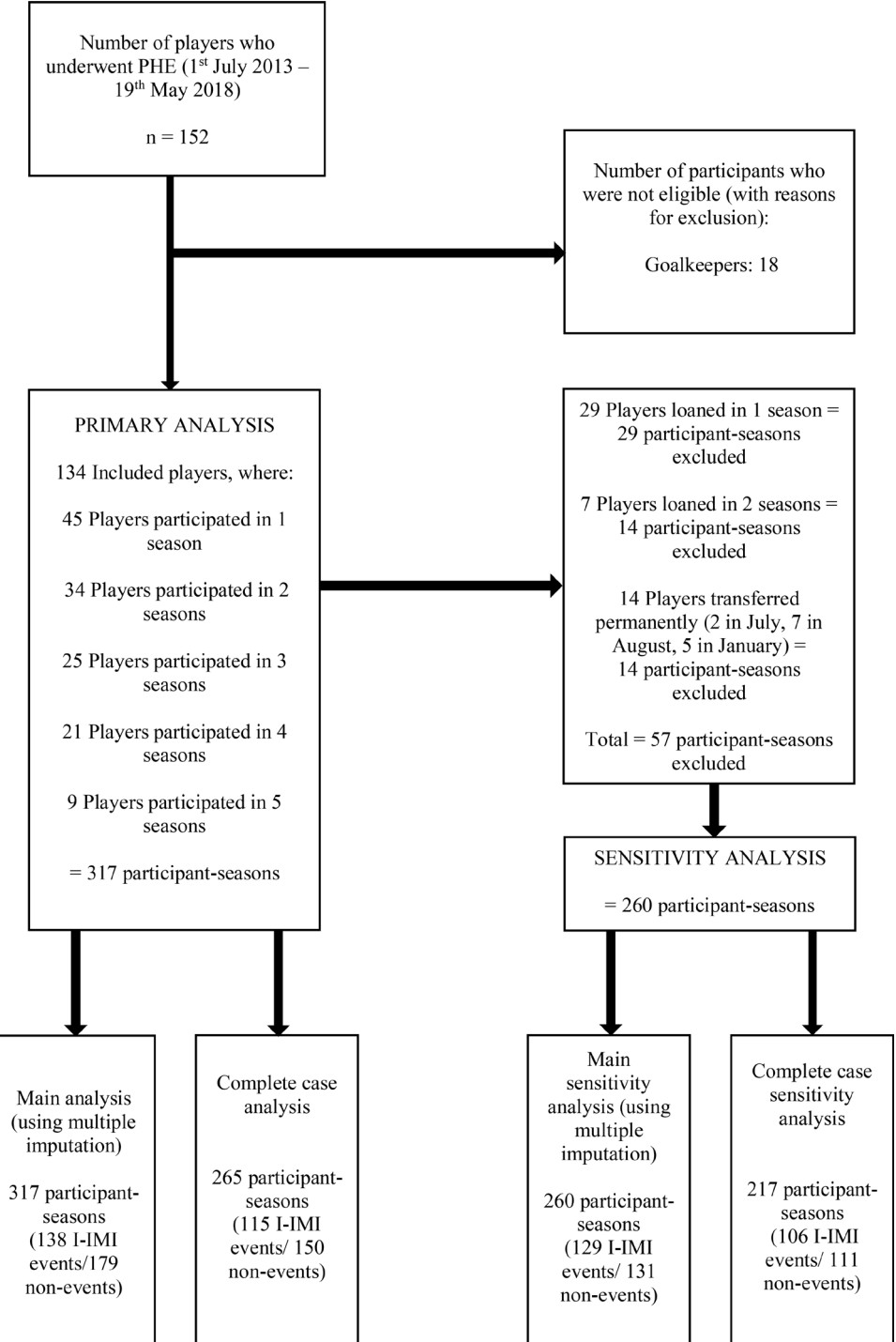

**Figure 1** Participant flow chart. I-IMI, index indirect muscle injury; n, number of participants.

p=0.02). However, no other candidates demonstrated prognostic importance. For most variables, the magnitude of the adjusted prognostic association was also smaller than the unadjusted association and some CIs were very wide.

## Complete-case and sensitivity analysis

The results of all complete-case and sensitivity analyses are presented in online supplemental files 6–8. Online supplemental file 9 shows a forest plot of the estimates obtained for all statistically significant candidate prognostic factors across all primary and sensitivity univariable analyses. Online supplemental file 10 shows a forest plot of the estimates obtained for all statistically significant candidate prognostic factors across all multivariable analyses.

For both univariable and multivariable analyses, the prognostic associations were very similar for the complete case and imputation analyses. Sensitivity analyses (ie,

**Table 3** Characteristics of included participants

| Characteristics/candidate variables | | Min | Lower quartile | Median | Upper quartile | Max. | Freq. (%)—if categorical | Missing values n (%) |
|---|---|---|---|---|---|---|---|---|
| Anthropometrics | | | | | | | | |
| 1. Age at PHE (years) | | 16.01 | 17.80 | 19.69 | 23.56 | 39.59 | – | 0 (0) |
| 2. Height (cm) | | 164.3 | 176.0 | 180.0 | 185.5 | 195.0 | – | 18 (5.68) |
| 3. Weight (kg) | | 56.8 | 69.2 | 73.6 | 80.0 | 94.0 | – | 18 (5.68) |
| Past medical history | | | | | | | | |
| Within 3 years prior to PHE, freq. of: | | | | | | | | |
| 4. Foot/ankle injuries | | 0 | 0 | 1 | 2 | 7 | – | 0 (0) |
| 5. Hip/groin injuries | | 0 | 0 | 0 | 1 | 5 | – | 0 (0) |
| 6. Knee injuries | | 0 | 0 | 0 | 1 | 3 | – | 0 (0) |
| 7. Shoulder injuries | | 0 | 0 | 0 | 0 | 2 | – | 0 (0) |
| 8. Lumbar spine injuries | | 0 | 0 | 0 | 0 | 3 | – | 0 (0) |
| 9. Iliopsoas IMIs | | 0 | 0 | 0 | 0 | 2 | – | 0 (0) |
| 10. Hip adductor IMIs | | 0 | 0 | 0 | 0 | 3 | – | 0 (0) |
| 11. Hamstring IMIs | | 0 | 0 | 0 | 1 | 6 | – | 0 (0) |
| 12. Quadriceps IMIs | | 0 | 0 | 0 | 0 | 3 | – | 0 (0) |
| 13. Calf IMIs | | 0 | 0 | 0 | 0 | 4 | – | 0 (0) |
| Within 3 years prior to PHE, most recent: | | | | | | | | |
| 14. Foot/ankle injury | Never | – | – | – | – | – | 143 (45.11) | 0 (0) |
| | <6 months | – | – | – | – | – | 43 (13.56) | 0 (0) |
| | 6–12 months | – | – | – | – | – | 34 (10.73) | 0 (0) |
| | >12 months | – | – | – | – | – | 97 (30.60) | 0 (0) |
| 15. Hip/groin injury | Never | – | – | – | – | – | 217 (68.45) | 0 (0) |
| | <6 months | – | – | – | – | – | 23 (7.26) | 0 (0) |
| | 6–12 months | – | – | – | – | – | 23 (7.26) | 0 (0) |
| | >12 months | – | – | – | – | – | 54 (17.03) | 0 (0) |
| 16. Knee injury | Never | – | – | – | – | – | 201 (63.41) | 0 (0) |
| | <6 months | – | – | – | – | – | 15 (4.73) | 0 (0) |
| | 6–12 months | – | – | – | – | – | 31 (9.78) | 0 (0) |
| | >12 months | – | – | – | – | – | 70 (22.08) | 0 (0) |
| 17. Shoulder injury | Never | – | – | – | – | – | 297 (93.69) | 0 (0) |
| | <6 months | – | – | – | – | – | 6 (1.89) | 0 (0) |
| | 6–12 months | – | – | – | – | – | 4 (1.26) | 0 (0) |
| | >12 months | – | – | – | – | – | 10 (3.15) | 0 (0) |
| 18. Lumbar spine injury | Never | – | – | – | – | – | 264 (83.28) | 0 (0) |
| | <6 months | – | – | – | – | – | 8 (2.52) | 0 (0) |
| | 6–12 months | – | – | – | – | – | 9 (2.84) | 0 (0) |
| | >12 months | – | – | – | – | – | 36 (11.36) | 0 (0) |
| 19. Iliopsoas IMI | Never | – | – | – | – | – | 287 (90.54) | 0 (0) |
| | <6 months | – | – | – | – | – | 2 (0.63) | 0 (0) |
| | 6–12 months | – | – | – | – | – | 9 (2.84) | 0 (0) |
| | >12 months | – | – | – | – | – | 19 (5.99) | 0 (0) |
| 20. Hip adductor IMI | Never | – | – | – | – | – | 263 (82.92) | 0 (0) |
| | <6 months | – | – | – | – | – | 18 (5.68) | 0 (0) |
| | 6–12 months | – | – | – | – | – | 12 (3.79) | 0 (0) |
| | >12 months | – | – | – | – | – | 24 (7.57) | 0 (0) |

Continued

**Table 3** Continued

| Characteristics/candidate variables | | Min | Lower quartile | Median | Upper quartile | Max. | Freq. (%)—if categorical | Missing values n (%) |
|---|---|---|---|---|---|---|---|---|
| 21. Hamstring IMI | Never | – | – | – | – | – | 231 (72.87) | 0 (0) |
| | <6 months | – | – | – | – | – | 11 (3.47) | 0 (0) |
| | 6–12 months | – | – | – | – | – | 30 (9.46) | 0 (0) |
| | >12 months | – | – | – | – | – | 45 (14.20) | 0 (0) |
| 22. Quadriceps IMI | Never | – | – | – | – | – | 267 (84.23) | 0 (0) |
| | <6 months | – | – | – | – | – | 7 (2.21) | 0 (0) |
| | 6–12 months | – | – | – | – | – | 13 (4.10) | 0 (0) |
| | >12 months | – | – | – | – | – | 30 (9.46) | 0 (0) |
| 23. Calf IMI | Never | – | – | – | – | – | 283 (89.27) | 0 (0) |
| | <6 months | – | – | – | – | – | 11 (3.47) | 0 (0) |
| | 6–12 months | – | – | – | – | – | 6 (1.89) | 0 (0) |
| | >12 months | – | – | – | – | – | 17 (5.36) | 0 (0) |
| Musculoskeletal examination | | | | | | | | |
| 24. Mean PROM hip IR (deg.) | | 9.5 | 22.5 | 33.0 | 40.0 | 55.0 | – | 20 (6.31) |
| 25. Mean PROM hip ER (deg.) | | 17.5 | 33.5 | 38.5 | 43.0 | 62.0 | – | 20 (6.31) |
| 26. Mean hip flexor length (deg.) | | –7.0 | 3.5 | 9.0 | 15.0 | 55.0 | – | 23 (7.26) |
| 27. Mean hamstring/neural mobility length (deg.) | | 45.0 | 84.0 | 90.0 | 90.0 | 102.0 | – | 20 (6.31) |
| 28. Mean calf muscle length (deg.) | | 9.5 | 25.0 | 30.0 | 36.0 | 57.5 | – | 20 (6.31) |
| Strength/power tests | | | | | | | | |
| 29: Max. leg extension power difference (W/kg$^{-0.67}$) | | –11.94 | –1.55 | 0.29 | 1.77 | 15.26 | – | 45 (14.20) |
| 30: Mean of max. leg extension power (W/kg$^{-0.67}$) | | 23.01 | 42.12 | 46.52 | 51.95 | 78.69 | – | 45 (14.20) |
| 31: Max. leg extension velocity difference (m.s$^{-1}$) | | –0.34 | –0.05 | 0.02 | 0.11 | 0.40 | – | 41 (12.93) |
| 32: Mean of max. leg extension velocity (m.s$^{-1}$) | | 1.02 | 1.68 | 1.84 | 1.98 | 2.35 | – | 41 (12.93) |
| 33: Max leg extension force difference (N/kg$^{-0.67}$) | | –83.59 | –6.18 | 1.17 | 4.40 | 55.41 | – | 45 (14.20) |
| 34: Mean of max. leg extension force (N/kg$^{-0.67}$) | | 50.19 | 98.59 | 101.44 | 113.44 | 217.95 | – | 45 (14.20) |
| 35: CMJ force per kg of body mass | | 20.20 | 23.40 | 25.40 | 28.04 | 39.20 | – | 42 (13.25) |
| 36: CMJ height (cm) | | 28.7 | 37.3 | 40.2 | 43.0 | 58.0 | – | 42 (13.25) |

Note that for variables that state between limb differences (ie, variables 29, 31 and 33), positive values indicate greater left limb values compared with right limb values; negative values indicate greater right limb values compared to left limb values.
N (note: N/kg$^{-0.67}$ has a scaling factor to normalise force to body mass).[54] W (note: W/kg$^{-0.67}$ has a scaling factor to normalise force to body mass).[54]
CMJ, countermovement jump; deg., degrees; ER, external rotation; IMI, indirect muscle injury; IR, internal rotation; m.s$^{-1}$, metres per second; PHE, periodic health examination; PROM, passive range of movement.

excluding players who were loaned or transferred) led to slightly stronger prognostic associations for some factors (eg, the frequency and timing of previous hamstring injuries), although the CIs were wider.

## DISCUSSION

This exploratory study has examined linear and non-linear prognostic associations between PHE-derived variables and I-IMIs in elite football players, using routinely collected data from a five-season period.

While the univariable analyses suggested six candidate factors are associated with I-IMIs, such analyses are limited as they only provide baseline association measures prior to adjustment for other important prognostic factors.[26] However, after adjustment in the multivariable analyses, the evidence indicates most PHE derived variables did not add any prognostic value over and above age. The only exception was that if a hamstring IMI was sustained by a player more than 12 months (but less than 3 years) prior to PHE, their odds of sustaining a lower extremity I-IMI significantly increased 2.2-fold, which has not been

**Table 4** Univariable and multivariable logistic regression estimates for all candidate variables

| Candidate PF and type | Univariable (unadjusted) | | | | Multivariable (adjusted for age, height, weight) | | | |
|---|---|---|---|---|---|---|---|---|
| | OR | 95% CI | P value | Best model fit | OR | 95% CI | P value | Best model fit |
| Anthropometric (adjustment factors): | | | | | | | | |
| 1: Age (years) | **1.12** | **1.06 to 1.18** | **<0.001** | Linear | **1.12** | **1.05 to 1.18** | **<0.001** | Linear |
| 2: Height (cm) | 1.03 | 0.99 to 1.07 | 0.13 | – | 1.02 | 0.97 to 1.07 | 0.46 | – |
| 3: Weight (kg) | **1.03** | **1.00 to 1.07** | **0.03** | Linear | 1.00 | 0.95 to 1.04 | 0.91 | – |
| Within 3 years preceding PHE, frequency of: | | | | | | | | |
| 4: Foot/ankle injuries | 1.04 | 0.87 to 1.24 | 0.68 | – | 1.04 | 0.87 to 1.25 | 0.65 | – |
| 5: Hip/groin injuries | 1.16 | 0.90 to 1.51 | 0.25 | – | 1.29 | 0.99 to 1.70 | 0.06 | – |
| 6: Knee injuries | 0.96 | 0.72 to 1.29 | 0.81 | – | 0.98 | 0.72 to 1.32 | 0.88 | – |
| 7: Shoulder injuries | 2.38 | 0.98 to 5.75 | 0.05 | – | 1.89 | 0.76 to 4.74 | 0.17 | – |
| 8: Lumbar spine injuries | 0.88 | 0.60 to 1.29 | 0.50 | – | 1.08 | 0.72 to 1.61 | 0.71 | – |
| 9: Iliopsoas IMIs | 0.73 | 0.38 to 1.43 | 0.37 | – | 0.86 | 0.43 to 1.71 | 0.67 | – |
| 10: Hip adductor IMIs | 1.38 | 0.92 to 2.09 | 0.12 | – | 1.18 | 0.76 to 1.84 | 0.46 | – |
| 11: Hamstring IMIs | **1.56** | **1.17 to 2.09** | **<0.001** | Linear | 1.35 | 1.00 to 1.83 | 0.05 | – |
| 12: Quadriceps IMIs | 1.08 | 0.67 to 1.73 | 0.75 | – | 1.05 | 0.65 to 1.71 | 0.84 | – |
| 13: Calf IMIs | **1.80** | **1.09 to 2.97** | **0.02** | Linear | 1.30 | 0.75 to 2.25 | 0.35 | – |
| Within 3 years preceding PHE, most recent: | | | | | | | | |
| 14: Foot/ankle injury (never) | ref | ref | ref | – | ref | ref | ref | – |
| 14: Foot/ankle injury (<6 months) | 1.27 | 0.64 to 2.53 | 0.49 | – | 1.41 | 0.69 to 2.87 | 0.35 | – |
| 14: Foot/ankle injury (6–12 months) | 1.16 | 0.54 to 2.46 | 0.71 | – | 1.15 | 0.52 to 2.52 | 0.73 | – |
| 14: Foot/ankle injury (>12 months) | 1.27 | 0.75 to 2.13 | 0.37 | – | 1.22 | 0.71 to 2.09 | 0.47 | – |
| 15: Hip/groin injury (never) | ref | ref | ref | – | ref | ref | ref | – |
| 15: Hip/groin injury (<6 months) | 1.05 | 0.44 to 2.49 | 0.92 | – | 1.66 | 0.67 to 4.13 | 0.27 | – |
| 15: Hip/groin injury (6–12 months) | 0.59 | 0.23 to 1.50 | 0.27 | – | 0.65 | 0.25 to 1.69 | 0.38 | – |
| 15: Hip/groin injury (>12 months) | 1.58 | 0.87 to 2.87 | 0.14 | – | 1.80 | 0.96 to 3.36 | 0.07 | – |
| 16: Knee injury (never) | ref | ref | ref | – | ref | ref | ref | – |
| 16: Knee injury (<6 months) | 1.15 | 0.40 to 3.28 | 0.80 | – | 1.12 | 0.37 to 3.34 | 0.84 | – |
| 16: Knee injury (6–12 months) | 1.23 | 0.58 to 2.62 | 0.60 | – | 1.24 | 0.57 to 2.70 | 0.60 | – |
| 16: Knee injury (>12 months) | 0.93 | 0.53 to 1.61 | 0.79 | – | 0.99 | 0.56 to 1.75 | 0.96 | – |
| 17: Shoulder injury (never) | ref | ref | ref | – | ref | ref | ref | – |
| 17: Shoulder injury (<6 months) | 2.75 | 0.50 to 15.26 | 0.25 | – | 2.55 | 0.44 to 14.72 | 0.30 | – |
| 17: Shoulder injury (6–12 months) | 1.38 | 0.19 to 9.90 | 0.75 | – | 1.11 | 1.15 to 8.36 | 0.92 | – |

Continued

**Table 4** Continued

| Candidate PF and type | Univariable (unadjusted) | | | | Multivariable (adjusted for age, height, weight) | | | |
|---|---|---|---|---|---|---|---|---|
| | OR | 95% CI | P value | Best model fit | OR | 95% CI | P value | Best model fit |
| 17: Shoulder injury (>12 months) | 3.21 | 0.81 to 12.67 | 0.10 | – | 2.36 | 0.58 to 9.62 | 0.23 | – |
| 18: Lumbar spine injury (never) | ref | ref | ref | – | ref | ref | ref | – |
| 18: Lumbar spine injury (<6 months) | 1.24 | 0.30 to 5.05 | 0.77 | – | 2.10 | 0.50 to 8.82 | 0.31 | – |
| 18: Lumbar spine injury (6–12 months) | 0.62 | 0.15 to 2.53 | 0.50 | – | 0.75 | 0.18 to 3.14 | 0.69 | – |
| 18: Lumbar spine injury (>12 months) | 0.70 | 0.34 to 1.44 | 0.33 | – | 0.97 | 0.46 to 2.06 | 0.94 | – |
| 19: Iliopsoas IMI (never) | ref | ref | ref | – | ref | ref | ref | – |
| 19: Iliopsoas IMI (<6 months) | 1.24 | 0.77 to 20.05 | 0.88 | – | 1.27 | 0.08 to 21.32 | 0.87 | – |
| 19: Iliopsoas IMI (6–12 months) | 0.62 | 0.15 to 2.53 | 0.51 | – | 0.83 | 0.20 to 3.43 | 0.80 | – |
| 19: Iliopsoas IMI (>12 months) | 0.57 | 0.21 to 1.55 | 0.27 | – | 0.66 | 0.24 to 1.82 | 0.42 | – |
| 20: Hip adductor IMI (never) | ref | ref | ref | – | ref | ref | ref | |
| 20: Hip adductor IMI (<6 months) | 1.37 | 0.53 to 3.56 | 0.52 | – | 1.29 | 0.48 to 3.47 | 0.61 | |
| 20: Hip adductor IMI (6–12 months) | 1.37 | 0.43 to 4.36 | 0.60 | – | 1.16 | 0.35 to 3.81 | 0.81 | – |
| 20: Hip adductor IMI (>12 months) | 1.37 | 0.59 to 3.16 | 0.46 | – | 1.14 | 0.47 to 2.79 | 0.77 | – |
| 21: Hamstring IMI (never) | ref | ref | ref | – | ref | ref | ref | – |
| 21: Hamstring IMI (<6 months) | 2.84 | 0.81 to 9.99 | 0.10 | – | 2.07 | 0.57 to 7.56 | 0.27 | – |
| 21: Hamstring IMI (6–12 months) | 1.42 | 0.66 to 3.06 | 0.37 | – | 1.22 | 0.56 to 2.70 | 0.62 | – |
| 21: Hamstring IMI (>12 months) | **2.95** | **1.51 to 5.73** | **<0.001** | Linear | **2.24** | **1.11 to 4.53** | **0.02** | Linear |
| 22: Quadriceps IMI (never) | ref | ref | ref | – | ref | ref | ref | – |
| 22: Quadriceps IMI (<6 months) | 1.74 | 0.38 to 7.91 | 0.48 | – | 1.56 | 0.33 to 7.36 | 0.58 | – |
| 22: Quadriceps IMI (6–12 months) | 0.58 | 0.17 to 1.93 | 0.37 | – | 0.59 | 0.17 to 2.04 | 0.41 | – |
| 22: Quadriceps IMI (>12 months) | 1.14 | 0.53 o 2.43 | 0.74 | – | 1.07 | 0.49 to 2.35 | 0.86 | – |
| 23: Calf IMI (never) | ref | ref | ref | – | ref | ref | ref | – |
| 23: Calf IMI (<6 months) | 3.78 | 0.98 to 14.56 | 0.05 | – | 3.11 | 0.78 to 12.46 | 0.11 | – |
| 23: Calf IMI (6–12 months) | 7.09 | 0.82 to 61.51 | 0.08 | – | 3.80 | 0.41 to 35.41 | 0.24 | – |
| 23: Calf IMI (>12 months) | 1.26 | 0.47 to 3.36 | 0.64 | – | 0.73 | 0.24 to 2.15 | 0.56 | – |
| Musculoskeletal: | | | | | | | | |
| 24. Mean PROM hip IR (deg.) | **0.97** | **0.95 to 0.99** | **0.01** | Linear | 0.98 | 0.95 to 1.00 | 0.06 | – |
| 25. Mean PROM hip ER (deg.) | 0.97 | 0.95 to 1.00 | 0.09 | – | 0.99 | 0.96 to 1.02 | 0.53 | – |
| 26. Mean hip flexor length (deg.) | 1.01 | 0.98 to 1.04 | 0.46 | – | 1.01 | 0.99 to 1.04 | 0.32 | – |
| 27. Mean hamstring/neural mobility length (deg.) | 0.99 | 0.96 to 1.02 | 0.53 | – | 0.98 | 0.95 to 1.02 | 0.33 | – |
| 28. Mean calf muscle length (deg.) | 1.00 | 0.97 to 1.02 | 0.77 | – | 1.00 | 0.97 to 1.02 | 0.79 | – |

Continued

**Table 4** Continued

| Candidate PF and type | Univariable (unadjusted) | | | | Multivariable (adjusted for age, height, weight) | | | |
|---|---|---|---|---|---|---|---|---|
| | OR | 95% CI | P value | Best model fit | OR | 95% CI | P value | Best model fit |
| **Strength/power:** | | | | | | | | |
| 29: Max. leg extension power difference (W/kg$^{-0.67}$) | 0.99 | 0.92 to 1.07 | 0.84 | – | 0.99 | 0.91 to 1.06 | 0.71 | – |
| 30: Mean of max. leg extension power (W/kg$^{-0.67}$) | 1.02 | 0.99 to 1.05 | 0.24 | – | 1.01 | 0.97 to 1.04 | 0.73 | – |
| 31: Max. leg extension velocity difference (m.s$^{-1}$) | 2.15 | 0.31 to 14.88 | 0.44 | – | 2.83 | 0.38 to 21.31 | 0.31 | – |
| 32: Mean of max. leg extension velocity (m.s$^{-1}$) | 1.96 | 0.68 to 5.64 | 0.21 | – | 1.48 | 0.49 to 4.47 | 0.49 | – |
| 33: Max leg extension force difference (N/kg$^{-0.67}$) | 0.99 | 0.98 to 1.02 | 0.76 | – | 1.00 | 0.98 to 1.02 | 0.65 | – |
| 34: Mean of max. leg extension force (N/kg$^{-0.67}$) | 1.00 | 0.99 to 1.01 | 1.00 | – | 1.00 | 0.99 to 1.01 | 0.72 | – |
| 35: CMJ force per kg of body mass (N/kg) | 0.99 | 0.93 to 1.06 | 0.78 | – | 0.99 | 0.92 to 1.07 | 0.80 | – |
| 36: CMJ height (cm) | 1.03 | 0.98 to 1.08 | 0.27 | – | 0.72 | 0.96 to 1.07 | 0.63 | – |

Note: ORs are expressed per one-unit increase for all continuous factors, and according to category for all categorical factors; factors in bold indicate significance at the <0.05 level.
N (note: N/kg$^{-0.67}$ has a scaling factor to normalise force to body mass).[54] W (note: W/kg$^{-0.67}$ has a scaling factor to normalise force to body mass.)[54]
–, not applicable; CI, confidence interval; CMJ, countermovement jump; deg., degrees; ER, external rotation; Freq, frequency; I-IMI, index indirect muscle injury; IMI, indirect muscle injury; IR, internal rotation; max, maximum; m.s$^{-1}$, metres per second; N, newtons; OR, odds ratio; PHE, periodic health examination; PROM, passive range of movement; ref, reference category; SLR, straight leg raise; WBL, weight-bearing lunge.

previously reported. Although not directly comparable, earlier studies have also shown that a history of a previous hamstring IMI is specifically associated with an increased hazard of future hamstring IMIs in elite players.[14 17 19] Nevertheless, the uncertainty in our estimates (demonstrated by wide 95% CIs) and differences observed during the sensitivity analyses mean that this variable only has provisional prognostic value and needs to be established in further confirmatory studies.

Indeed, age was the only variable that could be considered as an important prognostic factor, which is easily obtained even without conducting PHE. For illustration, our estimates suggest that for every 1-year increase in age, the odds of sustaining an I-IMI during a season would increase by approximately 12%. As an example, to put into the context of absolute risk, for two players who were the same height and weight but aged 5 years apart, if the younger-aged player had a risk of 0.44 (which was overall outcome prevalence in our study), then the older player would have a risk of 0.58.

The findings of this study confirm those of a previous study that developed and validated a multivariable prognostic model to predict lower extremity IMI risk in elite football players using PHE data, where age was considered an important prognostic factor (OR 1.10, 95% CI 1.03 to 1.17).[13] Other studies have shown that age is a multivariable prognostic factor specifically associated with increased hamstring IMI risk (OR range 1.40–1.78),[14 15 20] although the reported estimates were larger than those observed in our study. These differences may be due to chance or partly because we merged all lower extremity I-IMI outcomes rather than using IMI subgroups, which may have diluted the strength of our observed associations. However, although our approach was less clinically meaningful, merging I-IMI outcomes was essential in order to maximise the statistical power of our study.

Importantly, while its prognostic importance has been confirmed in multiple studies[13–15 20] age is not a causal factor for future IMI occurrence. Rather, it is likely to be a proxy marker for another potential causal mechanism. Taking this and the non-modifiable nature of age into account, this factor could not be used clinically to inform specific injury mitigation interventions, so should only be considered useful to explain differences in risk between players in a team, or included in future prognostic model development studies.

Using data from PHE tests that measure modifiable physical and performance characteristics has been previously questioned for injury prediction purposes.[2] Our results fully support this view, because none of the modifiable musculoskeletal (clinical examination) or strength and power tests evaluated showed any statistically significant associations with I-IMIs. This absence of strong associations mean that such tests have poor discriminatory ability, usually because of overlap in test scores that occur in individuals who sustain a future injury and those who do not.[2] Furthermore, after measurement at a solitary timepoint (ie, preseason), it

is likely that the prognostic value of modifiable factors is time-varying[39] as a consequence of physical and physiological adaptations that occur from training exposure and other injuries.[40]

Overall, when considering the findings of this study and the related previous prognostic model development and validation study,[13] the majority of PHE derived candidate variables cannot be considered useful for IMI risk prediction and injury prevention practice in elite football players. However, because of this study's exploratory nature (with many estimates having very wide CIs), the shortcomings of the current evidence base and the paucity of known prognostic factors in elite football,[14] there is a clear need for further investigation in this area to improve our understanding of the prognostic value of PHE in elite football and other sports.

### Limitations and future research

This study is unique in that we have investigated non-linear associations as per methodological guidelines.[12 41] However, in the analyses, non-linear associations were not found to be superior to linear associations. For practical reasons, our imputation model did not assume non-linear associations and therefore, may have reduced the ability to detect genuine non-linear relationships in the subsequent analyses. However, this is not a concern for age, as there were no missing values for this factor and is unlikely to be a material concern for all other factors as missing data was always less than 15%.

A competing risks analysis was not conducted. This meant that individuals who sustained injury types other than lower extremity IMIs were still considered at risk, even though this may have affected their training and match exposure and hence their risk of sustaining an I-IMI event. Candidate factors were only measured at one timepoint each season, which means that dynamic associations were not investigated. We also assumed that participant-seasons were independent. Future studies could account for competing risks, use repeated measurements over time (ie, using intermittent PHE, conducted at various stages throughout the season) and incorporate between-season correlations into analyses. However, the complexity of such analyses would also require a significantly larger volume of data. This could be achieved through data sharing initiatives and individual participant data meta-analyses, which would also increase the power to detect genuine prognostic associations and non-linear relationships.

Finally, it is acknowledged that the PHE data used in this study and the related prognostic model development study was restricted to 60 candidate variables overall, from a limited selection of PHE procedures.[13 25] Further studies should investigate a wider selection of PHE tests, including (but not limited to) other musculoskeletal, biomechanical, imaging and other in vivo diagnostic tests for example, providing that the quality of data is robust through evaluation of reliability and validity.

## CONCLUSION

This study has evaluated prognostic associations between PHE-derived candidate variables and lower extremity I-IMIs in elite football players using data that were routinely collected over five seasons. No clear associations were found for nearly all PHE variables, although if a player sustained a hamstring IMI greater than 12 months (but less than 3 years) prior to PHE, then this had potential prognostic value over and above the prognostic value of age. Indeed, age was the only variable to be confirmed as a clear prognostic factor in both univariable and multivariable analyses. However, this is easily measured without the need to conduct PHE, and although it has limited application in clinical practice, it should be included as an important factor in any future prognostic model development studies. Overall, the PHE processes used in this study and the related prognostic model development study cannot be currently considered as a useful source of prognostic factors for I-IMI risk prediction and injury prevention practice in elite football players. Instead, they should only be considered potentially useful for screening of pathology, as well as for rehabilitation and performance monitoring. Further research is required to confirm the prognostic value of PHE for IMI risk prediction and to identify novel prognostic factors that could improve development of prognostic models in football and other sports.

**Author affiliations**
[1]Football Medicine and Science Department, Manchester United Football Club, Manchester, UK
[2]Centre for Epidemiology Versus Arthritis, University of Manchester, Manchester, UK
[3]Department of Health Professions, Manchester Metropolitan University, Manchester, UK
[4]Research Institute for Primary Care and Health Sciences, Keele University, Keele, UK
[5]Centre for Biostatistics, Manchester Academic Health Science Centre, Manchester, UK

**Contributors** TH was responsible for the conceptualisation of the project, study design, database construction, data extraction and cleaning, protocol development and protocol writing. TH conducted the data analysis, interpretation and wrote the main manuscript. RR provided statistical guidance and assisted with development of the study design, analysis and edited manuscript drafts. MJC assisted with the study conceptualisation and design, protocol development, clinical interpretation and editing the manuscript drafts. JCS provided guidance with the study design, development of the analysis and protocol, interpretation of the analysis, as well as editing the study manuscripts. All authors read and approved this final manuscript. TH is the guarantor, and accepts full responsibility for the finished work, the conduct of the study, has had access to the data and controlled the decision to publish. MJC and JCS are joint last authors.

**Funding** The lead researcher (TH) is receiving sponsorship from Manchester United Football Club to complete a postgraduate PhD study programme. This work was also supported by Versus Arthritis: grant number 21755.

**Competing interests** None declared.

**Patient and public involvement** Patients and/or the public were not involved in the design, or conduct, or reporting, or dissemination plans of this research.

**Patient consent for publication** Not applicable.

**Ethics approval** Informed consent was not required as data were captured from the mandatory PHE completed through the participants' employment. The data usage was approved by the club and the Research Ethics Service at the University of Manchester.

**Data availability statement** Data may be available upon reasonable request. An anonymised summary of the dataset that was analysed during this study may be available from the corresponding author on reasonable request.

**ORCID iDs**
Tom Hughes http://orcid.org/0000-0003-2266-6615
Richard Riley http://orcid.org/0000-0001-8699-0735

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
