## [Reviewer comments · BMJ Open]

ARTICLE DETAILS

TITLE (PROVISIONAL)	Can prognostic factors for indirect muscle injuries in elite football (soccer) players be identified using data from preseason screening? An exploratory analysis using routinely-collected periodic health examination records.
AUTHORS	Hughes, Tom; Riley, Richard; Callaghan, Michael J.; Sergeant, Jamie

VERSION 1 – REVIEW

REVIEWER	L. Falese University of Cassino and Southern Lazio
REVIEW RETURNED	07-Oct-2021

GENERAL COMMENTS	Thank you very much for the opportunity to review this interesting paper. I think that this research could prove to be very interesting and useful to a very large audience and it can be suitable for publication in BMJ Open after a minor revision. The article provides a clear title and abstract helping the reader to have a proper idea of the research question, aim and design of the study. However, since the paper is very long, sometimes during the reading, it can be easy to get lost or to find repetitions. The protocol used is relatively easy to be reproduced and it could be interesting to see if there would be similar results with a different sample. Here you can find more specific comments: INTRODUCTION The introduction provides a good literature background of the topic especially concerning the prognostic value of the PHE. However, less is commented about injuries, data and prevention strategies (only a short sentence at line 49 page 5 of 66 in the pdf file). After stating that PHE is important and why, the authors offer no clear explanation of why they chose the indirect injuries as the outcome of the present work. The authors may wish to add a small paragraph to better clarify why it is so important to understand predictor variables of indirect injuries according to the existing scientific literature. From line 49/50 page 5 of the pdf file, in order to make the motivation clearer, the authors described their previous findings that lead them carry out this research. In my opinion this paragraph should be synthesized (maybe it could be commented more in the discussion..) and they should refer to their previous article without writing “our review” “we developed” but in a more neutral way. METHODS The experimental apparatus is properly described and appropriate for the study. However, I have some small comment: Line 17 of page 8 of the pdf file: Even though the club is very well known, the authors should add that Manchester United Football
--

	Club is in UK. Line 33-36 page 7 of the pdf file: in my opinion the physical tests implemented should be shortly described and a reference of their validation should be added. Line 17 page 8 of the pdf file: Ethics. Maybe it would have been good to have a Privacy Policy stating that the data could have been used anonymously for research reasons. Table 2: in my opinion it is not relevant to put this table in the paper but it may be something that would be helpful to add as a supplementary material. RESULTS Table 3 and 4: I would delete the number written near to the name of the variable (1. Age; 2 height and so on). DISCUSSION The discussion is very well written and the findings are properly described in the context of the current literature. However, sometimes it sounds too repetitive. The authors may wish to avoid too many repetitions of concepts since the paper is already long. Finally, since the findings did not show statistically relevant associations, in my opinion, it may be worthwhile to discuss more detailed ideas for further investigation.
--	--

REVIEWER	Stephen McAleer Red Bull Performance
REVIEW RETURNED	25-Nov-2021

GENERAL COMMENTS	Overall impression: The paper is a good attempt to translate the practices within elite football into research and evaluate the effectiveness of the practice. It is an extremely challenging environment to conduct research within and reflects what is done by practitioners in most clubs. However the environment and elite population implicitly imposes limitations on the quality of the research. However research in this field is warranted and will make practitioners reflect more fully on the screening and PHE within sports teams. This piece has been described as an exploratory study by the authors and I think this reflects the limitations within the study. Overall the paper is well written and the authors should be applauded for their attempt to transfer applied work into research. Some key overall points that need addressing - The outcome is indirect lower limb muscle injuries. However the testing procedure has little testing done in relation to the main lower limb muscle injuries related to football - notably hamstrings, adductors and calves. The testing reported is insufficient to report that PHE cannot be considered a useful source of prognostic factors for IMI risk prediction. 2. The missing data reflects what conducting research is like within this environment. However including this data restricts the quality and impact of the work. It leads to confusing and convoluted reading, particularly within the methods. The authors could possibly eliminate this data and go only with the complete cases. Personally I like cleaner data but the exploratory nature of the study may lend it to be included. 3. The authors have gone to extensive lengths with data and statistical analysis and this is to be applauded. Abstract injury 'prediction' is possibly not a term I greatly enjoy personally. However I appreciate it is used within the literature and is accepted
---

terminology. Injury risk / mitigation / prognostic model / reduction / influence etc.

Missing data line within abstract and detracts from the abstract on first reading

'over linear ones' - possibly change to 'over linear associations'

Background

Page 4 line 15-17 - are these references in support this statement or in what context are they being used

Page 4 line 17-22 - refs 2 & 4 again are slightly conflicting to support the statement. Review and consider which refs are relevant / supportive of each statement

page 4 second paragraph - opening sentence does not read well, consider revising

Methods section - May be worth defining variable and candidate definition as used throughout study and can be confusing to read particularly in results section

Consider impact of using the missing data.

Page 8 PHE variables - please provide more clarity on the inclusion and exclusion of variables and parameters. It is slightly confusing to read

Table 1 & candidate variables - The paper looks at I-IMIs in lower abdominal, hip, thighs, calf or foot. However the variables chosen have only one measure say for hamstring, (mobility/neural length). The only strength measure recorded were variations on leg extension. Fairly big limitation in the study. Can you explain why these or strength variables were not included? Would inform the reader better

Page 10 re Missing Data. I am getting this right that missing data were given normal scores? Please clarify further

Results:

Generally the results section reflects the overall study aims and is clear to read.

Page 14 Table 3 - the figures seem very approximate. e.g. the min, median etc seem strange for mean figures whilst using an inclinometer?

Page 18 complete case section - I think this is of more interest to the reader. The 'cleaner' data set is more easily understood and read. Including the missing data adds confusion and potentially detracts from the study. I understand if the authors want to use that data for statistical power but it detracts from the quality of the study. However it does highlight 'real world' data collection difficulties within an elite population and setting.

Discussion:

Probably the section that needs most work. The implications derived from the results are far too wide reaching given the limited nature of the methodology in relation to muscle injuries.

Page 19 Majority of findings have been on hamstring IMI. Would have been very interesting if more hamstring related testing particularly strength / biomech measures were incorporated into PHE.

Page 20 22-30 - yes this increased the statistical power but it is certainly a limitation. If the PHE was more specific to each subgroup

	for injuries associated with football (calf, hamstring, quadriceps, adductors etc) then IMI subgroups would have been appropriate and interesting, whether statistically significant or not. Page 21 - lines 5-10 A relevant point. However the issue with the modifiable MSK, strength or power tests evaluated is the limited scope of the tests used. Essentially one leg extension exercise and a CMJ is the strength and power assessments. Lines 11-17 - too broad and sweeping statement given the limitations of the pHE used. Lines 17-22 - is it worthwhile more intermittent screening within a season rather than just pre season? Worth a discussion point in your discussion
--	--

VERSION 1 – AUTHOR RESPONSE

Reviewer: 1

Dr. L. Falese, University of Cassino and Southern Lazio Comments to the Author:

Thank you very much for the opportunity to review this interesting paper.

I think that this research could prove to be very interesting and useful to a very large audience and it can be suitable for publication in BMJ Open after a minor revision.

The article provides a clear title and abstract helping the reader to have a proper idea of the research question, aim and design of the study. However, since the paper is very long, sometimes during the reading, it can be easy to get lost or to find repetitions.

The protocol used is relatively easy to be reproduced and it could be interesting to see if there would be similar results with a different sample.

Thank you for taking the time to read our paper, and also for your helpful comments.

Here you can find more specific comments:

INTRODUCTION

The introduction provides a good literature background of the topic especially concerning the prognostic value of the PHE. However, less is commented about injuries, data and prevention strategies (only a short sentence at line 49 page 5 of 66 in the pdf file). After stating that PHE is important and why, the authors offer no clear explanation of why they chose the indirect injuries as the outcome of the present work. The authors may wish to add a small paragraph to better clarify why it is so important to understand predictor variables of indirect injuries according to the existing scientific literature.

From line 49/50 page 5 of the pdf file, in order to make the motivation clearer, the authors described their previous findings that lead them carry out this research. In my opinion this paragraph should be synthesized (maybe it could be commented more in the discussion..) and they should refer to their previous article without writing “our review” “we developed” but in a more neutral way.

Thank you for bringing these points to our attention, and we apologise if the reason for selecting indirect muscle injuries was unclear. We have revised the introduction and placed more emphasis on why indirect muscle injuries have been selected. Furthermore, we have also rephrased any reference to our previous work in a more neutral way in both the

introduction and discussion. The revised introduction can now be located on pages 4-6 and the excerpt is also found below.

“BACKGROUND

Periodic health examination (PHE), or screening, is a well-established clinical evaluation strategy in elite football.¹ Typically during PHE, players undertake various medical, musculoskeletal, functional and performance tests² during preseason and in-season periods.¹ PHE allows opportunities for general health surveillance, identification of salient pathology³ and monitoring of rehabilitation or performance.⁴ In addition, although PHE cannot establish specific causal factors for injuries,⁴ it is perceived to be useful for the prediction of future injury risk in athletes,^{2,4} which could prove especially valuable for injury types that are problematic in terms of incidence and severity. Indirect (non-contact) muscle injuries (IMIs) are an obvious example, because they account for between 30.3% to 47.9% of all injuries observed in elite football⁵⁻⁹ and each IMI typically results in 14.4⁵ to 15 days lost to training and competition.⁸

To be able to predict the probability of the occurrence of future health events, prognostic factors are required.¹⁰ In the context of football, prognostic factors could be any PHE derived variables, characteristics or measurements (e.g. medical history, leg strength or range of motion tests) that are associated with increased injury risk through causal or non-causal pathways.⁴ There is clinical value in gaining a deeper understanding of prognostic factors associated with injuries such as IMIs. Specifically, prognostic factors can help practitioners understand the differences in risk (outcome event probability) between players, and therefore explain why some players may have a better or worse prognosis than others.⁴ Furthermore, prognostic factors that have an established causal role in injury occurrence can inform the selection of injury mitigation strategies, relevant for subgroups of players who share such characteristics.⁴ Finally, causal prognostic factors can also be used to develop innovative intervention approaches aimed at mitigating risk.¹⁰

Despite these benefits, the predictive power of single prognostic factors are limited.^{11,12} However, if several prognostic factors are used in combination within a multivariable prognostic model, it may be possible to produce useful individualised risk estimates^{10,11} that can be used to communicate risks to players, practitioners and coaches.¹³ Additionally, if developed using prognostic factors which have a causal role in injury risk, prognostic models could also be used to assist practitioners in selecting an array of specific risk reduction interventions that are bespoke to the prognostic factor profile of individual players⁴.

Because the predictive function of PHE remains unsubstantiated^{3,14} and given that indirect (non-contact) muscle injuries (IMIs) are the most significant problem observed in elite football,⁵⁻⁹ a related multivariable prognostic model was recently developed to predict individualised lower extremity IMI risk in elite players using PHE data.¹³ However, sample size limitations meant that only 10 candidate prognostic factors could be considered in the model and these were selected using data quality assessment, clinical reasoning, or on the basis of a

related systematic review.¹⁴ The performance of the model was modest and it was concluded that implementing it in practice would not be beneficial.¹³

Furthermore, several methodological limitations of the current evidence have been previously highlighted, which specifically included inadequate reporting of outcomes, prognostic factor measurement and reliability.¹⁴ Additionally, while most studies performed appropriate statistical analyses, continuous prognostic factor measurements were often categorised¹⁵⁻¹⁸ and non-linear associations were not investigated,¹⁵⁻²¹ which does not conform to current methodological recommendations.²²⁻²⁴

To further the development of IMI prognostic models and improve understanding of how differences in IMI risk may occur between individuals, there is a clear need to ascertain the existence of other robust and novel prognostic factors.¹³ Therefore, this study used routinely collected data from a 5-season period to explore; 1) prognostic associations between PHE-derived data and IMI outcomes in elite footballers, using a broader dataset than had been considered in the development the previous prognostic model¹³ and; 2) the prognostic value of these PHE-derived data over and above standard anthropometric data, including age (which has previously confirmed prognostic value^{13 14}), height and weight. Both linear and non-linear associations were also explored which, as far as is known, has not been conducted previously.”

METHODS

The experimental apparatus is properly described and appropriate for the study. However, I have some small comment.

Line 17 of page 8 of the pdf file: Even though the club is very well known, the authors should add that Manchester United Football Club is in UK.

Thank you for highlighting this point. Unfortunately, since we submitted this paper, we have had instruction from the legal team at Manchester United that we should not refer to the football club directly by name within the methods, although we are able to continue to state our affiliation. Consequently, we have had to revise the statement, which can now be located on Page 6, paragraph 3 and reads:

“Eligible participants were identified from a population of male elite footballers, aged 16-40 years old at an English Premier League football club.”

Line 33-36 page 7 of the pdf file: in my opinion the physical tests implemented should be shortly described and a reference of their validation should be added.

Thank you for your suggestion. We apologise for the lack of description of the tests and this was an oversight on our part. In our companion paper, we attached the description of all the tests as a supplementary file. Because of the length of the paper, we have elected to do the same in this paper. Please refer to Supplementary file 1 for this addition. This has been highlighted to the reader also, on Page 7, paragraph 2:

“A description of all of the included test procedures is presented in Supplementary File 1.”

Furthermore, as requested, we have now added a column to Table 1 highlighting the validity with supported references (where possible.) The revised table can be found on page 10 and also is shown below.

Table 1: List of candidate variables with a summary of the units of measurement, methods and reliability of measurement and data type.

Candidate Variable Type	Name of candidate variable	Candidate variable Identification number	Measurement unit	Measurement Method	Reliability (if applicable/available)	Validity (if applicable/available)	Data type	
Anthropometric	Age	1	Years	Medical records	-	-	Cont.	
	Height	2	Centimetres (cm)	Standing height measure	-	-	Cont.	
	Weight	3	kilograms (kg)	Digital scales	-	-	Cont.	
Medical history	Within 3 years prior to PHE, the frequency of:	Foot/ankle injuries	4	Count	Medical records	-	-	Dis./cont.
		Hip/groin injuries	5	Count	Medical records	-	-	Dis./cont.
		Knee injuries	6	Count	Medical records	-	-	Dis./cont.
		Shoulder injuries	7	Count	Medical records	-	-	Dis./cont.
		Lumbar spine injuries	8	Count	Medical records	-	-	Dis./cont.
		Iliopsoas	9	Count	Medical records	-	-	Dis./cont.

	injuries							.
	Hip adductor IMIs	10	Count	Medical records	-	-		Dis./cont .
	Hamstring IMIs	11	Count	Medical records	-	-		Dis./cont .
	Quadriceps IMIs	12	Count	Medical records	-	-		Dis./cont .
	Calf IMIs	13	Count	Medical records	-	-		Dis./cont .
Within 3 years prior to PHE, the most recent	Foot/ankle injury	14	Never, < 6 months, 6-12 months, > 12 months.	Medical records	-	-		Cat.
	Hip/groin injury	15	Never, < 6 months, 6-12 months, > 12 months.	Medical records	-	-		Cat.
	Knee injury	16	Never, < 6 months, 6-12 months, > 12 months.	Medical records	-	-		Cat.
	Shoulder injury	17	Never, < 6 months, 6-12 months, > 12 months.	Medical records	-	-		Cat.

			months.				
Lumbar spine injury	18	Never, < 6 months, 6-12 months, > 12 months.	Medical records	-	-	Cat.	
Iliopsoas injury	19	Never, < 6 months, 6-12 months, > 12 months.	Medical records	-	-	Cat.	
Hip adductor IMI	20	Never, < 6 months, 6-12 months, > 12 months.	Medical records	-	-	Cat.	
Hamstring IMI	21	Never, < 6 months, 6-12 months, > 12 months.	Medical records	-	-	Cat.	
Quadriceps IMI	22	Never, < 6 months, 6-12 months, > 12 months.	Medical records	-	-	Cat.	
Calf IMI	23	Never, < 6 months, 6-12 months, > 12 months.	Medical records	-	-	Cat.	

Musculoskeletal	Mean PROM hip internal	24	Degrees	Digital inclinometer + ROM	Intra-rater ICC= 0.90 ³⁴	-	Cont.
------------------------	----	---------	----------------------------	--	---	-------

	rotation (IR)						
	Mean PROM hip external rotation (ER)	25	Degrees	Digital inclinometer + ROM	Intra-rater ICC = 0.90 ³⁴	-	Cont.
	Mean hip flexor length	26	Degrees	Digital inclinometer + Thomas Test	Inter-rater ICC = 0.89 ³⁵	Concurrent validity with handheld goniometer (r=0.86-0.92) ³⁵ and IMC (r=0.49-0.53) ³⁶	Cont.
	Mean hamstring length/neural mobility	27	Degrees	Digital inclinometer + SLR	Intra-rater ICC = 0.95-0.98 ³⁷ Inter-rater ICC = 0.80-0.97 ³⁷	Construct validity with handheld inclinometer (r=0.98-0.99) ³⁸	Cont.
	Mean calf muscle length	28	Degrees	Digital inclinometer + WBL	Inter-rater ICC = 0.80- 0.95 ^{39 40} Intra-rater ICC = 0.88 ⁴⁰	Concurrent validity of inclinometer with 2D motion analysis (r range=0.71-0.76) ⁴¹	Cont.
Strength/power	Max. leg extension power difference	29	Normalised watts per kilo (W/kg ^{-0.67})	Double leg press using Keiser Air 300 machine	Test-retest ICC = 0.886 ⁴²	Concurrent validity with mounted force plate (r=0.952) ⁴³	Cont.

Mean of max. leg extension power	30	Normalised watts per kilo ($W/kg^{-0.67}$)	Double leg press using Keiser Air 300 machine	Test-retest ICC = 0.886 ⁴²	Concurrent validity with mounted force plate ($r=0.977$) ⁴³	Cont.
Max. leg extension velocity difference	31	Peak velocity ($m.s^{-1}$)	Double leg press using Keiser Air 300 machine	Test-retest ICC = 0.792 ⁴²	Concurrent validity with mounted force plate ($r=0.999$) ⁴³	Cont.
Mean of max. leg extension velocity	32	Peak velocity ($m.s^{-1}$)	Double leg press using Keiser Air 300 machine	Test-retest ICC = 0.792 ⁴²	Concurrent validity with mounted force plate ($r=0.999$) ⁴³	Cont.
Max leg extension force difference	33	Normalised peak force ($N/Kg^{-0.67}$)	Double leg press using Keiser Air 300 machine	Test-retest ICC = 0.914 ⁴²	Concurrent validity with mounted force plate ($r=0.994$) ⁴³	Cont.
Mean of max. leg extension force	34	Normalised peak force ($N/Kg^{-0.67}$)	Double leg press using Keiser Air 300 machine	Test-retest ICC = 0.914 ⁴²	Concurrent validity with mounted force plate ($r=0.994$) ⁴³	Cont.
CMJ Force per kg of body mass	35	Force per kg (N/kg)	Countermovement jump (CMJ) + force plate	Test-retest ICC = 0.80-0.88 ⁴⁴	Concurrent validity with other force plate ($r \geq 0.99$) ⁴⁵	Cont.
CMJ height	36	Centimetres (cm)	Countermovement jump (CMJ) + force plate	Test-retest ICC = 0.80-0.88 ⁴⁴	Concurrent validity with other force plate ($r \geq 0.99$) ⁴⁵	Cont.

Key: BMI= body mass index; cat.= categorical; Cont.=continuous; cm = centimetres; CMJ=countermovement jump; deg. = degrees; dis./cont.= discrete treated as continuous; grp=group; ICC=intraclass correlation coefficient; I-IMI= index indirect muscle injury; IMC= inertial motion capture; Kg=kilograms; kg/m² = kilograms/body height squared; max.=maximum; m.s.= metres per second; N= newtons (note: N/kg^{-0.67} has a scaling factor to normalise force to body mass ⁴⁶); PROM=passive range of movement; SLR= straight leg raise; W= watts (note: W/kg^{-0.67} has a scaling factor to normalise force to body mass ⁴⁶); WBL=weight bearing lunge.

Line 17 page 8 of the pdf file: Ethics. Maybe it would have been good to have a Privacy Policy stating that the data could have been used anonymously for research reasons.

Thank you for your comment. Indeed, we had stated this already at the end of the manuscript according to the required 'Data sharing statement'. This can be located on page 26 where it states:

DATA SHARING STATEMENT

An anonymised summary of the dataset that was analysed during this study may be available from the corresponding author on reasonable request.

Table 2: in my opinion it is not relevant to put this table in the paper but it may be something that would be helpful to add as a supplementary material.

Thank you for your comment. While for a more clinically orientated audience, we agree that this table may not immediately appear as useful as the other tables presented in the main body of the manuscript, the Reporting Recommendations for Marker Prognostic Studies (ReMARK) (recommended for reporting prognostic factor research), states that the full reporting all statistical analyses is a key aspect that should be presented (Sauerbrei et al, 2018). Consequently, because of the emphasis on reporting these in the guidelines, we feel that the table should remain located in the main text.

Sauerbrei, W., Taube, S.E., McShane, L.M., Cavenagh, M. (2018) Reporting Recommendations for Tumour Marker Prognostic Studies (ReMARK), Journal of the National Cancer Institute, 110 (8), pp.803-811.

RESULTS

Table 3 and 4: I would delete the number written near to the name of the variable (1. Age; 2 height and so on).

Thank you for your comment and acknowledge that your suggestion may help to simplify the table. However, the numbers in these tables are for identification purposes and also correspond to the variables in Tables 1 and 2. As such, we have opted to keep the numbers in the table to maintain consistency across the paper.

DISCUSSION

The discussion is very well written and the findings are properly described in the context of the current

literature. However, sometimes it sounds too repetitive. The authors may wish to avoid too many repetitions of concepts since the paper is already long. Finally, since the findings did not show statistically relevant associations, in my opinion, it may be worthwhile to discuss more detailed ideas for further investigation.

Thank you for your feedback regarding the discussion. After reviewing it again following your feedback, we can understand your point of view, and consequently we have revised it accordingly to try to reduce the repetition (especially related to the prognostic factor of age). The revised sections of the manuscript can be found on pages 21-24, and now states:

“This exploratory study has examined linear and non-linear prognostic associations between PHE-derived variables and I-IMIs in elite football players, using routinely collected data from a 5-season period.

While the univariable analyses suggested 6 candidate factors are associated with I-IMIs, such analyses are limited as they only provide baseline association measures prior to adjustment for other important prognostic factors.²⁶ However, after adjustment in the multivariable analyses, the evidence indicates most PHE derived variables did not add any clear prognostic value over and above age. The only exception was that if a hamstring IMI was sustained by a player more than 12 months (but less than 3 years) prior to PHE, their odds of sustaining a lower extremity I-IMI significantly increased 2.2-fold, which has not been previously reported. Although not directly comparable, earlier studies have also shown that a history of a previous hamstring IMI is specifically associated with an increased hazard of future hamstring IMIs in elite players.^{14 17 19} Nevertheless, the uncertainty in our estimates (demonstrated by wide 95% confidence intervals) and differences observed during the sensitivity analyses mean that this variable only has provisional prognostic value and needs to be established in further confirmatory studies.

Indeed, age was the only variable that could be considered as an important prognostic factor, which is easily obtained even without conducting PHE. For illustration, our estimates suggest that for every 1-year increase in age, the odds of sustaining an I-IMI during a season would increase by approximately 12%. As an example, to put into the context of absolute risk, for two players who were the same height and weight but aged 5 years apart, if the younger-aged player had a risk of 0.44 (which was overall outcome prevalence in our study), then the older player would have a risk of 0.58.

The findings of this present study confirm those of a previous study that developed and validated a multivariable prognostic model to predict general lower extremity IMI risk in elite football players using PHE data, where age was considered an important prognostic factor (OR = 1.10, 95% CI = 1.03 to 1.17).¹³ Other studies have shown that age is a multivariable prognostic factor specifically associated with increased hamstring IMI risk (OR range 1.40-1.78),^{14 15 20} although the reported estimates were larger than those observed in our study. These differences may be due to chance or partly because we merged all lower extremity I-IMI outcomes rather than utilising IMI subgroups, which may have diluted the strength of our

observed associations. However, although our approach was less clinically meaningful, merging I-IMI outcomes was essential in order to maximise the statistical power of our study.

Importantly, while its prognostic importance has been confirmed in multiple studies^{13-15 20} age is not a causal factor for future IMI occurrence. Rather, it is likely to be a proxy marker for another potential causal mechanism. Taking this and the non-modifiable nature of age into account, this factor could not be used clinically to inform specific injury mitigation interventions, so should only be considered useful to explain differences in risk between players in a team, or included in future prognostic model development studies.

Using data from PHE tests that measure modifiable physical and performance characteristics has been previously questioned for injury prediction purposes.² Our results fully support this view, because none of the modifiable musculoskeletal (clinical examination) or strength and power tests evaluated showed any statistically significant associations with I-IMIs. This absence of strong associations mean that such tests have poor discriminatory ability, usually because of overlap in test scores that occur in individuals who sustain a future injury and those who do not.² Furthermore, after measurement at a solitary timepoint (i.e. pre-season), it is likely that the prognostic value of modifiable factors is time-varying⁵² as a consequence of physical and physiological adaptations that occur from training exposure and other injuries.⁵³

Overall, when considering the findings of this study and the related previous prognostic model development and validation study,¹³ the majority of PHE derived candidate prognostic factors cannot currently be considered useful for IMI risk prediction and injury prevention practice in elite football players. However, because of this study's exploratory nature (with many estimates having very wide confidence intervals), the shortcomings of the current evidence base and the paucity of known prognostic factors in elite football,¹⁴ there is a clear need for further investigation in this area to improve our understanding of the prognostic value of PHE in elite football and other sports.

Limitations and future research

This study is unique in that we have investigated non-linear associations as per methodological guidelines.^{12 54} However, in the analyses, non-linear associations were not found to be superior to linear associations. For practical reasons, our imputation model did not assume non-linear associations and therefore may have reduced the ability to detect genuine non-linear relationships in the subsequent analyses. However, this is not a concern for age, as there were no missing values for this factor and is unlikely to be a material concern for all other factors as missing data was always less than 15%.

A competing risks analysis was not conducted, which meant that individuals who sustained injury types other than lower extremity IMIs were still considered at risk, even though this may have affected their training and match exposure and hence risk of sustaining an I-IMI event. Candidate factors were only measured at one timepoint each season, which means that dynamic associations were not investigated. We also assumed that participant-seasons were independent. Future studies could account for competing risks, utilise repeated

measurements over time (i.e. using intermittent PHE, conducted at various stages throughout the season) and incorporate between-season correlations into analyses. However, the complexity such analyses would also require a significantly larger volume of data. This could be achieved through data sharing initiatives and individual participant data meta-analysis, which would also increase the power to detect genuine prognostic associations and non-linear relationships.

Finally, it is acknowledged that the PHE data used in this study and the related prognostic model development study was restricted to 60 candidate prognostic factors overall, from a limited selection of PHE procedures.^{13 25} Further studies should investigate a wider selection of PHE tests, including (but not limited to) other musculoskeletal, biomechanical, imaging and other in vivo diagnostic tests for example, providing that the quality of data is robust through evaluation of reliability and validity.”

Reviewer: 2

Dr. Stephen McAleer, Red Bull Performance Comments to the Author:

Overall impression:

The paper is a good attempt to translate the practices within elite football into research and evaluate the effectiveness of the practice. It is an extremely challenging environment to conduct research within and reflects what is done by practitioners in most clubs. However the environment and elite population implicitly imposes limitations on the quality of the research. However research in this field is warranted and will make practitioners reflect more fully on the screening and PHE within sports teams. This piece has been described as an exploratory study by the authors and I think this reflects the limitations within the study.

Overall the paper is well written and the authors should be applauded for their attempt to transfer applied work into research.

Thank you for your positive and constructive feedback. We feel that your comments have certainly helped us to strengthen the paper.

1. Some key overall points that need addressing - The outcome is indirect lower limb muscle injuries. However the testing procedure has little testing done in relation to the main lower limb muscle injuries related to football - notably hamstrings, adductors and calves. The testing reported is insufficient to report that PHE cannot be considered a useful source of prognostic factors for IMI risk prediction.

Thank you for your comment and for bringing our attention to this point. This statement was made to in relation to the results of the present paper in combination with our related previous model development study (which utilised other relevant PHE tests such as previous muscle injury history, hip range of movement, hamstring muscle length, countermovement jump etc); see Hughes et al, 2020), but we agree that this was not made clear in the original text. Consequently, we have modified the phrase to be more specific, which can be located on Page 23, paragraph 2 and now states:

Overall, when considering the findings of this study and the related previous prognostic model development and validation study,¹³ the majority of PHE derived candidate prognostic factors cannot be considered useful for IMI risk prediction and injury prevention practice in elite football players.

This has also been amended in the conclusion, which can be found on page 25.

“Overall, the PHE processes used in this study and the related prognostic model development and validation study cannot be currently considered as a useful source of prognostic factors for I-IMI risk prediction and injury prevention practice in elite football players. Instead, they should only be considered potentially useful for screening of pathology, as well as rehabilitation and performance monitoring.”

Reference:

Hughes, T et al (2020) The Value of Preseason Screening for Injury Prediction: The Development and Internal Validation of a Multivariable Prognostic Model to Predict Indirect Muscle Injury Risk in Elite Football (Soccer) Players. *Sports Medicine - Open* 2020;6(22) doi: <https://doi.org/10.1186/s40798-020-00249-8>

2. The missing data reflects what conducting research is like within this environment. However including this data restricts the quality and impact of the work. It leads to confusing and convoluted reading, particularly within the methods. The authors could possibly eliminate this data and go only with the complete cases. Personally I like cleaner data but the exploratory nature of the study may lend it to be included.

Thank you for your suggestion. While we agree that missing data is common and is reflective of the practical difficulties in conducting research in an applied sports medicine setting, unfortunately omitting data from any analyses of prognostic factor or prognostic model development and validation studies is not recommended for several reasons. Firstly, if a complete case analysis is performed in isolation, this is both inefficient because as you say,

this reduces the available sample, but it also wastes other remaining data that may be useful in the analysis (Moons et al, 2015). Possibly more importantly, omitting participants with incomplete data could bias the results if the remaining participants are not representative of the entire study sample (Moons et al, 2015). This was indeed the case, because our missing data analysis (performed in our protocol), highlighted that those with missing observations were generally older, taller and heavier than those without and were therefore mainly representative of senior/first team players.

When there are systematic differences between participants with and without missing data (such as observed in our study) and these differences do not depend on the values of the missing data itself but rather on other observed variables, this is referred to as the data being missing at random (Sterne et al, 2009). In these circumstances, multiple imputation is recommended; this is where multiple copies of the dataset are created and missing values are replaced by imputed values that are sampled from the predicted distribution, and is based on the existing (observed) data (Sterne et al, 2009; Moons et al, 2015). Any models created are then fitted to the multiple datasets, and estimates are averaged across all datasets, allowing the uncertainty in the missing values to be reflected.

Because this is the recommended approach for prognostic research (Riley et al, 2019), we feel that it is best to maintain the analyses as they are presented (i.e. using multiple imputation) as this reduces potential bias, and allowed us to utilise a larger sample, thus maximising the statistical power of our study.

References:

Moons KG, Altman DG, Reitsma JB, et al. Transparent Reporting of a multivariable prediction model for Individual Prognosis or Diagnosis (TRIPOD): explanation and elaboration. *Ann Intern Med* 2015;162(1):W1-73. doi: 10.7326/M14-0698 [published Online First: 2015/01/07]

Riley RD, van der Windt DA, Croft P, Moons K.G. *Prognosis Research in Healthcare: Concepts, Methods and Impact*, Oxford University Press; Oxford.

Sterne et al (2009) Multiple imputation for missing data in epidemiological and clinical research; potential pitfalls, *British Medical Journal*, 338, b2393.

3. The authors have gone to extensive lengths with data and statistical analysis and this is to be applauded.

Thank you for your feedback.

Abstract

injury 'prediction' is possibly not a term I greatly enjoy personally. However I appreciate it is used within the literature and is accepted terminology. Injury risk / mitigation / prognostic model / reduction / influence etc.

Missing data line within abstract and detracts from the abstract on first reading 'over linear ones' - possibly change to 'over linear associations'

Thank you for pointing this out and we agree that changing the existing terminology makes the statement clearer. This has been amended as requested and now states on page 2:

“Allowing non-linear associations conferred no advantage over linear associations.”

Background

Page 4 line 15-17 - are these references in support this statement or in what context are they being used
Page 4 line 17-22 - refs 2 & 4 again are slightly conflicting to support the statement. Review and consider which refs are relevant / supportive of each statement

Thank you for your suggestion. Indeed Bahr’s (2016) paper primarily covers why, in his opinion, PHE is unable to be used for injury prediction purposes. In contrast, the paper by Hughes et al (2018) argues that it is theoretically possible to use PHE for this purpose if used around a prognosis research framework. However, both papers are consistent in that they suggest that it is *perceived* to be useful for prediction of injury. Consequently, we feel that this statement is referenced appropriately.

page 4 second paragraph - opening sentence does not read well, consider revising

We appreciate you highlighting this issue and apologise for the lack of clarity. This sentence has now been revised and can be found on page 4, which now states:

“To be able to predict the risk of future health events, prognostic factors are required.¹⁰ In the context of football, prognostic factors could be any PHE derived variables, characteristics or measurements (e.g. medical history, leg strength or range of motion tests) that are associated with increased injury risk through causal or non-causal pathways.⁴”

Methods section - May be worth defining variable and candidate definition as used throughout study

and can be confusing to read particularly in results section Consider impact of using the missing data.

Again, we apologise for the lack of clarity in related to this terminology. In the introduction the aim has now been rephrased, to avoid initial confusion, which on page 6 now states:

“To further the development of IMI prognostic models and improve understanding of how differences in IMI risk may occur between individuals, there is a clear need to ascertain the existence of other robust and novel prognostic factors.⁹ Therefore, this study used routinely collected data from a 5-season period to explore; 1) prognostic associations between PHE-derived data and IMI outcomes in elite footballers, using a broader dataset than had been considered in the development the previous prognostic model⁹ and; 2) the prognostic value of these PHE-derived data over and above standard anthropometric data, including age (which has previously confirmed prognostic value^{9 10}), height and weight. Both linear and non-linear associations were also explored which, as far as is known, has not been conducted previously.”

As requested, a small paragraph that defines ‘prognostic factor’ and ‘candidate variable’ has now also been added later in the Methods, under the PHE-derived Candidate Variables subheading (page 9) and states:

“To aid clarity in this study, the term ‘prognostic factor’ is reserved for factors found to have a prognostic association with an I-IMI outcome (i.e. with statistical evidence established during the analyses), whereas the term ‘candidate variables’ relates to all factors for which the association with I-IMI outcome was investigated during the analyses.”

Page 8 PHE variables - please provide more clarity on the inclusion and exclusion of variables and parameters. It is slightly confusing to read Table 1 & candidate variables - The paper looks at I-IMIs in lower abdominal, hip, thighs, calf or foot. However the variables chosen have only one measure say for hamstring, (mobility/neural length). The only strength measure recorded were variations on leg extension. Fairly big limitation in the study. Can you explain why these or strength variables were not included? Would inform the reader better Page 10 re Missing Data.

Thank you for your feedback. Because of the retrospective nature of the study we were restricted to data that were available, whilst also removing data that were not deemed high quality because of issues with the volume of missing data, reliability etc. We have outlined the process that we followed for the inclusion of the variables (i.e. those included in our related paper, and those selected for this paper) on page 9. However, we did not refer the reader to our study protocol. We have now amended this text, which can be found on page 9, where it now states:

“As described in the study protocol, the dataset contained 60 variables²⁶ that were eligible for analysis unless there were >15% missing observations or if reliability (where applicable) was reported as fair to poor (that is, Intraclass Correlation Coefficient (ICC) < 0.70).^{26,33} If any variables did not meet these eligibility criteria, they were excluded (Supplementary File 2). Collinearity between eligible variables was assessed with a scatterplot matrix; this was evident when tests were used to measure right and left limbs independently.²⁶ In these circumstances, composite variables were created for between-limb differences and the mean of the test measurements for both limbs, as described in the study protocol.²⁶”

Of the remaining eligible variables, 10 were used in a previous study to develop a multivariable prognostic model for I-IMI prediction (represented by 12 parameters).⁹ With the exception of age at PHE (which was used for adjustment purposes in this study), these candidates were therefore excluded.²⁶ The final number of candidate variables included for exploratory analysis was 36.”

I am getting this right that missing data were given normal scores? Please clarify further

In terms of your question, the included data that were continuous in nature demonstrated a non-normal distribution. To utilise multiple imputation using a multivariate normal distribution approach (also known as joint modelling), these data need to be transformed to approximate normality before being incorporated into the multiple imputation model. Using ‘normal scores’ is a method of approximating normality by ranking the data and calculating normal quantiles for the dataset. This was done automatically using the ‘Normal scores’ Stata code, referenced within the paper. Following imputation, analysis took place on the original scales of the variables.

Results:

Generally the results section reflects the overall study aims and is clear to read.

Page 14 Table 3 - the figures seem very approximate. e.g. the min, median etc seem strange for mean figures whilst using an inclinometer?

Thank you for your question, which is also related to our response above. If data have a non-normal distribution (as was the case for several continuous variables in this study), using the mean was not an appropriate method to describe the data. In such cases, using the median, range and quartiles are a more appropriate way to present the data (Bland, 2015), hence why we chose to present them within Table 3.

Reference:

Bland, M (2015) An Introduction to Medical Statistics, 4th Ed. Oxford.

Page 18 complete case section - I think this is of more interest to the reader. The 'cleaner' data set is more easily understood and read. Including the missing data adds confusion and potentially detracts from the study. I understand if the authors want to use that data for statistical power but it detracts from the quality of the study. However it does highlight 'real world' data collection difficulties within an elite population and setting.

Please refer to our elaboration earlier within our response that justifies our use of multiple imputation to handle missing data as the more methodologically-sound approach. The complete case analysis is included for comparison.

Discussion:

Probably the section that needs most work. The implications derived from the results are far too wide reaching given the limited nature of the methodology in relation to muscle injuries.

Page 19 Majority of findings have been on hamstring IMI. Would have been very interesting if more hamstring related testing particularly strength / biomech measures were incorporated into PHE.

Thank you for suggestion. We agree that in the present study, the battery of tests that we included were limited; we were restricted to the available data given the retrospective nature of the study. Additionally, the available prognostic factors were reduced because some were already used in our previous model development study (the current study searched for prognostic associations beyond the variables included in the prognostic model development study), and we also chose to restrict the other candidate factors to only those recorded with the highest quality data.

We have now acknowledged this as a limitation in the Limitations and Future Research section, which can be found on Page XX. The excerpt is stated below:

“Finally, it is acknowledged that the PHE data used in this study and the related prognostic model development study was restricted to 60 candidate prognostic factors overall, from a limited selection of PHE procedures.^{13 25} Further studies should investigate a wider selection of PHE tests, including (but not limited to) other musculoskeletal, biomechanical, imaging and other in vivo diagnostic tests for example, providing that the quality of data is robust through evaluation of reliability and validity.”

Page 20 22-30 - yes this increased the statistical power but it is certainly a limitation. If the PHE was more specific to each subgroup for injuries associated with football (calf, hamstring, quadriceps, adductors etc) then IMI subgroups would have been appropriate and interesting, whether statistically significant or not.

Thank you for this observation. However, as stated in our text, even with the sample size that we were restricted to (i.e. 317 participant-seasons with 138 lower extremity I-IMI events), we conservatively estimated that this would only allow 80% power to detect an adjusted odds ratio of at least 1.6 for every one standard deviation increase in each candidate prognostic factor of interest. The injury data that were available were cleaned and placed into a database, and when we had analysed the frequency of injuries in each subgroup, this ranged from 36 for hamstring injuries to 2 for hip abductors (Hughes, 2020). Unfortunately, these numbers are simply too small to obtain precise estimates of association, and a concern was that the present study could suffer from false negative results due to lack of power, hence the reason why we chose not to investigate subgroups.

Reference:

Hughes, T (2020) The Value of Periodic Health Examination for Injury Prediction in Elite Football: Is the Prognosis Good?, PhD Thesis, University of Manchester, https://www.research.manchester.ac.uk/portal/files/173360188/FULL_TEXT.PDF

Page 21 - lines 5-10 A relevant point. However the issue with the modifiable MSK, strength or power tests evaluated is the limited scope of the tests used. Essentially one leg extension exercise and a CMJ is the strength and power assessments.

Thank you for bringing this to our attention. As stated earlier, we agree that in the present study, the battery of tests that we included were limited. We have also added in the paragraph to the Limitations and Future Research section as stated in the related comment above.

Lines 11-17 - too broad and sweeping statement given the limitations of the PHE used.

Thank you for your comment and for bringing our attention to this point. We have addressed this now as one of the first main points you suggested above and amended phrases in both the discussion and conclusion. For reference, the text has been restated below:

This section can be found on Page 23, paragraph 2 and now states:

Overall, when considering the findings of this study and the related previous prognostic model development and validation study,¹³ the majority of PHE derived candidate prognostic factors cannot currently be considered useful for IMI risk prediction and injury prevention practice in elite football players.

This has also been amended in the conclusion, which can be found on page 25.

“Overall, the PHE processes used in this study and the related prognostic model development and validation study cannot be currently considered as a useful source of prognostic factors for I-IMI risk prediction and injury prevention practice in elite football players. Instead, they should only be considered potentially useful for screening of pathology, as well as rehabilitation and performance monitoring.”

Lines 17-22 - is it worthwhile more intermittent screening within a season rather than just pre season? Worth a discussion point in your discussion.

Thank you for your feedback, and we apologise if this was not clear. We did refer to this in the Limitations and Future Research section, but to aid clarity we have added a short description contained within parentheses. This can be located on page 24, paragraph 2 and now states:

“Future studies could account for competing risks, utilise repeated measurements over time (i.e. using intermittent PHE, conducted at various stages throughout the season) and incorporate between-season correlations into analyses.”

VERSION 2 – REVIEW

REVIEWER	L. Falese University of Cassino and Southern Lazio
REVIEW RETURNED	10-Feb-2022
GENERAL COMMENTS	Dear Authors, thank you very much for revising your paper and taking into consideration my comments. I believe that the article is now suitable for publication.